# Speech Perception Improvement Algorithm Based on a Dual-Path Long Short-Term Memory Network

**DOI:** 10.3390/bioengineering10111325

**Published:** 2023-11-16

**Authors:** Hyeong Il Koh, Sungdae Na, Myoung Nam Kim

**Affiliations:** 1Department of Medical & Biological Engineering, Graduate School, Kyungpook National University, Daegu 41944, Republic of Korea; khi3001@naver.com; 2Department of Biomedical Engineering, Kyungpook National University Hospital, Daegu 41944, Republic of Korea; 3Department of Biomedical Engineering, School of Medicine, Kyungpook National University, Daegu 41944, Republic of Korea

**Keywords:** speech enhancement, STFT, LSTM, encoder–decoder structure, dual-path network, spectral extension block, mel-filter banks, merge algorithm

## Abstract

Current deep learning-based speech enhancement methods focus on enhancing the time–frequency representation of the signal. However, conventional methods can lead to speech damage due to resolution mismatch problems that emphasize only specific information in the time or frequency domain. To address these challenges, this paper introduces a speech enhancement model designed with a dual-path structure that identifies key speech characteristics in both the time and time–frequency domains. Specifically, the time path aims to model semantic features hidden in the waveform, while the time–frequency path attempts to compensate for the spectral details via a spectral extension block. These two paths enhance temporal and spectral features via mask functions modeled as LSTM, respectively, offering a comprehensive approach to speech enhancement. Experimental results show that the proposed dual-path LSTM network consistently outperforms conventional single-domain speech enhancement methods in terms of speech quality and intelligibility.

## 1. Introduction

Speech recognition is attracting attention as a promising technology for human–computer interaction (HCI) due to its various advantages, such as fast input/output speed. The natural and intuitive characteristics of speech enable faster multitasking and information input, which facilitates more effective processing for many tasks; however, maintaining the quality of speech signals becomes challenging in noisy environments and leads to information loss. Speech enhancement research aims to address this by removing noise and distortions from speech signals based on various audio signal processing techniques to continuously improve the overall intelligibility and perceived quality of speech. Speech enhancement is a preprocessing technique used in various fields, such as smart cars, smart hearing aids, and voice over internet protocol services [1,2]. Conventional speech enhancement algorithms rely primarily on the statistical modeling of either speech or noise, frequently yielding unpredictable performance outcomes in non-stationary noise scenarios [3,4,5].

In recent years, the use of deep learning techniques has become predominant in the field of speech enhancement. Mapping techniques that directly estimate clean components of input signals or features and masking techniques that indirectly remove noise components by estimating mask functions are commonly employed in supervised learning. These processes utilize a variety of network models, including feedforward neural networks (FNN), recurrent neural networks (RNN), and convolutional neural networks (CNN), and in recent years, long short-term memory (LSTM) has shown significant performance improvements over other neural networks [6,7,8,9]. Many deep learning-based speech enhancement methods utilize the magnitude values of the spectrum extracted via short-time Fourier transforms (STFT) as input data. The magnitude spectrum, which possesses regular characteristics, is not only easy to learn but also effective in distinguishing speech and noise components. However, this approach only improves the magnitude spectrum and does not deal with phases, leading to performance limitations. Two methods are proposed to address the phase estimation problem: one based on time analysis and another on time–frequency analysis in complex domains.

Luo et al. proposed a time domain method called a fully convolutional time domain audio separation network (Conv-TasNet) composed of 1D convolutional layers [10]. This model extracts time domain features without frequency decomposition, using convolutional layers, and estimates the ideal ratio mask separating noise and speech to enhance speech quality [11]; however, Conv-TasNet does not consider frequency components; minimizing distortions in the time domain does not guarantee the accuracy of speech spectrum estimation.

Hu et al. introduced a time–frequency domain method called the deep complex convolution current network (DCCRN). This structure is composed of complex convolutional layers and complex LSTM [12]. The network models noise and speech as a complex ratio mask using a complex neural network, capturing the correlation between the spectral information, representing frequency, and the phase information, representing time in a complex domain. This approach displays excellent noise reduction performance, even in low signal-to-noise ratio (SNR) environments, but successful noise reduction from the time–frequency representation requires high-resolution frequency decomposition of the noisy signal, which requires a long temporal window for the computation of the STFT. Longer windows result in a decrease in time resolution. This means that the ability to capture and process rapid changes in the speech signal is diminished.

To overcome these limitations, this paper proposes a dual-path LSTM network aimed at effectively capturing and exploiting distinctive speech characteristics across two distinct domains (time and time–frequency). These two paths utilize LSTM networks to map mask functions and enhance temporal and spectral features, respectively. To improve the semantic information of temporal features, a 1D convolutional encoder–decoder structure with non-negative constraints is incorporated in the time domain path, and a spectral extension block is introduced in the time–frequency domain path to preserve spectral details. Subsequently, a merge algorithm is applied to combine the enhanced features from each domain, promoting mutual supplementation and ultimately improving the quality of the speech signal. Experimental results demonstrate that the proposed dual-path LSTM network is competitive over conventional single-domain speech enhancement methods. The main contributions of this paper are as follows:
For more accurate speech restoration, a dual-path LSTM network is proposed that complementarily learns time and time–frequency domain information.A 1D convolutional encoder–decoder structure with non-negative constraints to improve semantic information about temporal features and a spectral extension block to complement spectral details was found to capture better acoustic information.Experimental results show that combinations of different processing domains can achieve better performance than single domains.

## 2. Materials and Methods

### 2.1. Datasets

The training dataset and test dataset consisted of audio data from various datasets released at INTER SPEECH 2021 for the Deep Noise Suppression (DNS) challenge. The speech data used for training was a subset of the Librispeech Corpus, which consists of over 500 h of recorded data from more than 2150 speakers [13]. The noise data included portions from the Audioset Corpus, Freesound, and WHAMR Corpus for a total of 180 h of data spread across 150 distinct categories [14,15]. The selected speech and noise data were mixed to create a total of 200 h of training noise–mixed data using audio synthesis scripts provided in the DNS challenge. To account for various noise environments, the mixed data was adjusted to contain random SNR values ranging from −5 to 25 dB by adjusting the levels of speech and noise during synthesis.

The 200 h training dataset was divided into separate sets for training (160 h) and cross-validation (40 h). Additionally, a separate test dataset was constructed to rigorously evaluate the model’s performance. The speech test dataset consisted of the Voice Bank Corpus of 28 speakers’ voices, while the noise test dataset was composed of the DEMAND Single Speaker and Noise Test Set, containing two types of artificial noise and eight types of natural noise scenarios. The test dataset was designed to simulate challenging noise environments commonly encountered in daily life, with SNRs ranging from 0 to 20 dB, and included a total of 2000 audio test examples [16,17]. The total number of data instances used was 22,000, distributed in an approximate ratio of 7:2:1 for training, validation, and testing, respectively. All data samples were formatted using a 16 kHz sampling frequency and saved in the WAV audio format.

### 2.2. Overview of the Proposed Method

The proposed dual-path LSTM-based speech enhancement network consists of three stages: a learnable encoder–decoder structure; a spectral extension block; and a merge module. The overall architecture of the algorithm is illustrated in Figure 1.

As shown in Figure 1, noisy input is divided into two pathways that process high-resolution temporal and spectral features. Segmentation and learnable encoders were used in the time path, and STFT and Mel-filter banks were used in the frequency path. These features model an ideal ratio mask that suppresses noise over an LSTM network. Subsequently, the noisy features are element-wise multiplied by the estimated ideal ratio mask to output only the enhanced speech component. The spectrum extension block compensates for missing details of the estimated spectrum by exchanging full-band and sub-band information of enhanced magnitude and Mel-spectral features. The compensated spectrum is restored into a waveform using the Griffin–Lim Algorithm (GLA), which reconstructs the phase information [18]. Restored with ISTFT (inverse STFT), the waveform is converted to a time domain representation via the reuse of the encoder’s 1D convolution layer to achieve dimensionality consistency with the temporal feature values estimated in the first path. After that, the calculated feature vectors are merged following the proposed merge algorithm. The merged speech feature vector is transformed into frame data via the decoder and synthesized into an enhanced speech signal.

### 2.3. Dual-Path LSTM-Based Speech Enhancement Model

#### 2.3.1. Speech Enhancement in the Time Domain

Speech typically possesses strong temporal structures, making it desirable and necessary for systems to model the time-dependent characteristics of noise and speech to enhance speech performance. In this context, a 1D convolution-based encoder–decoder deep learning framework was employed to achieve this goal.

Figure 2 illustrates the deep learning framework for the time domain proposed in this paper.

As shown in Figure 2, the noisy speech is segmented into frames of the form $X_{t}=[x_{1},\ldots \;x_{k},\ldots ,x_{T}]$ for detailed temporal structure analysis, where k represents the frame index, L is the window length, and T is the total number of frames. The segmented frames are then processed by the learnable weight of the 1D convolution filter in the encoder and converted into time domain features that represent the structural characteristics of the waveform. The visualization of the encoder output in Figure 2 shows that the dark and light colors of speech and noise are well distinguished. This indicates that the representation of the encoder is effective in speech estimation.

Equation (1) represents the output results of the encoder:(1)$$W=Q\left(X_{t}⊛U\right),$$
where *W* represents the output of the encoder; *Q* is a non-negative activation function; *U* represents the encoder filter’s weight functions; and ⊛ denotes the convolution operation. As per Equation (1), following the 1D convolution operation of *U*, the input segment is transformed into an N-dimensional representation corresponding to the length of the convolution filters and outputs only positive values based on the non-negative activation function. These constraints assume that the encoder output’s masking operation is only valid when noise and speech waveforms can be represented as combinations of non-negative weight functions. The unconstrained representation of the encoder can then be followed by an unconstrained mask, which can affect the results of the masking operation.

LSTM estimates a mask with values ranging from 0 to 1 based on the encoder output to distinguish noise from speech. The estimated mask extracts only the clean speech component via its operations with the encoded feature vector. The clean speech vector is then restored to its original input signal form via the decoder.

Equation (2) represents the output results of the decoder:(2)$$\hat{X_{t}}=\left(M⊙W\right)⊛Y,$$
where $\hat{X_{t}}$ represents the segment of the estimated clean speech signal from the network; *M* is the estimated mask from the network; *Y* represents the basic functions of the decoder filters; and ⊙ denotes the Hadamard product operation. As shown in Equation (2), the decoder uses a 1D transposed convolution operation that can be represented via matrix multiplication based on the representation generated in the encoder. These operations reconstruct the time features into a frame form and use an overlap-add (OLA) algorithm to synthesize them into a complete waveform [19].

#### 2.3.2. Speech Enhancement in the Frequency Domain

The frequency representation of a signal, known as the spectrum, is important for preserving the overall frequency structure of speech and its naturalness; however, such full-band information might be inadequate for removing specific types of noise present in certain frequency ranges. To address this limitation, a spectral extension block was employed.

Typically, the frequency range between 2 and 5 kHz is considered crucial for distinguishing and removing noise from speech. This range contains important bands for speech perception, including consonants and various formant information that determines speech intelligibility and quality, but it can also be challenging to remove noisy elements with irregular patterns, such as pink noise and other environmental noise, since these can overlap with the speech signal due to their abnormal patterns. Given that the Mel-spectrum accurately reflects the nonlinear frequency information perceived by the human auditory system, extracting the Mel-spectrum within a narrow frequency band emerges as a potent strategy for precise noise reduction in the spectral region most impactful on perceived speech quality. Additionally, the Mel-spectrum is less sensitive to small changes in frequency, thus ensuring robust performance against variations in noise signals. In this context, a method is proposed for effectively modeling complex spectral patterns inherent in various environmental noises by appropriately utilizing the sub-band Mel-spectrum based on human auditory characteristics to learn the important frequency structure of speech and the overall frequency structure of the full-band spectrum.

To capture detailed frequency dependencies, the input of the network in the second pathway transformed into the STFT form was represented as $X\left(t,f\right)=Mage^{i\theta }$, where *Mag* represents magnitude, $e^{i\theta }$ denotes phase spectrum, and *T* and *F* represent time and frequency resolutions, respectively. The magnitude spectrum is transformed into a sub-band Mel-spectrum $MS\left(t,f\right)$ covering the range of approximately 2000 to 5000 Hz via Mel-filter banks composed of m nonlinear filters and is concatenated with the full-band spectrum before being input to the LSTM network.

Equations (3)–(5) represent the frequency domain input information of the network:(3)$$mel\left(m,f\right)=\left\{\begin{array}{c} \;\;\;\;\;\;\;\;\;\;\;\;\;\;\;\;\;\;\;\;\;\;\;\;\;\;\;\;\;\;\;\;\;\;\;\;\;\;\;\;\;\;\;\;\;\;\;\;\;\;\;\;\;\;\;\;\;0,\;f<f_{mel}[m-1]\\ \frac{2(f-f_{mel}\left[f-m\right])}{\left(f_{mel}\left[m+1\right]-f_{mel}\left[m-1\right])(f_{mel}\left[m\right]-f_{mel}\left[m-1\right]\right)},\;f_{mel}[m-1]\leq f<f_{mel}[m]\\ \frac{2(f-f_{mel}\left[f-m\right])}{\left(f_{mel}\left[m+1\right]-f_{mel}\left[m-1\right])(f_{mel}\left[m\right]-f_{mel}\left[m-1\right]\right)},\;f_{mel}[m]\leq f<f_{mel}[m+1]\\ \;\;\;\;\;\;\;\;\;\;\;\;\;\;\;\;\;\;\;\;\;\;\;\;\;\;\;\;\;\;\;\;\;\;\;\;\;\;\;\;0,\;f\geq f_{mel}[m+1] \end{array}\right.,$$
(4)$$f_{mel}=2595\log_{10}\left(1+\frac{f}{700}\right),$$
(5)$$X_{f}=CONCAT\left(Mag\left(t,f\right),\;MS\left(t,f\right)\right),\;where\;MS\left(t,f\right)=\sum_{f=0}^{F-1}mel(m,f),$$
where $mel\left(m,f\right)$ represents the Mel-filter banks; $X_{f}$ is the concatenated magnitude spectrum; and CONCAT denotes the concatenation operation. As per Equation (5), the reason for connecting the spectrums is that when sub-band spectrums are concatenated and input as a single time step, the LSTM network can learn feature representations of human speech perception. This can help the network better capture the overall noise distribution in the sub-bands. The concatenated spectrums are then output in the form of a clean spectrum using the estimated LSTM mask. Figure 3 illustrates the configuration of the spectrum extension block, which integrates the advantages of the different frequency bands of the spectrum.

As shown in Figure 3, the concatenated sub-band Mel-spectrum and full-band spectrum are enhanced by computation of the LSTM with the estimated noise suppression mask and divided back into the Mel-spectrum and magnitude spectrum via a split function. The divided spectrums of the different frequency bands represent the magnitude spectrum of the signals with noise components removed. The Mel-spectrum is restored to the same dimension as the full-band spectrum via a pseudo-inverse matrix operation, and the improved components of the spectrum are compared via subtraction between the two spectrums to extract a more refined speech signal. Additionally, the Rectified Linear Unit (ReLU) function was employed to extract the spectral details that the sub-band Mel-spectrum was not able to estimate. The extracted speech component was operated on with the sub-band spectrum to yield a more accurate single speech spectrum. Then, the clean speech spectrum was combined with the noisy phase spectrum and transformed into the time domain spectrum using inverse STFT. This process can result in an imbalance between the magnitude and phase spectrums, which potentially limits the resulting audio quality. To address this, a reconstruction process aligning the phase to the spectrum was conducted using the Griffin–Lim Algorithm.

#### 2.3.3. Dual-Path Time–Frequency Merging Technique

In this section, the time–frequency domain merging algorithm is described, which integrates estimated speech features based on the parallel network proposed earlier, with the aim of maximizing the performance of the speech enhancement model.

The time and frequency features of the estimated speech from the previous step might not effectively remove noise in certain domain-specific parts or could have inaccurately removed portions. To estimate a more accurate speech component, a method was employed that involves dimension normalization and utilizing intersections, addressing potential discrepancies between different domain estimates.

Equations (6)–(8) illustrate the process of the proposed time–frequency domain feature merging algorithm.
(6)$$S_{t}={Mask}_{t}⊙Q\left(X_{t}⊛U\right),$$
(7)$$S_{f}=GLA\left({Mask}_{f}⊙X_{f}\right)⊛U,$$
(8)$$S_{merge}={(S}_{t}-ReLU(S_{t}-S_{f}))⊛Y,$$
where $S_{t}$ represents the time domain feature vector of the masked speech from the first pathway, $S_{f}$ signifies the value reshaped via the encoder based on the masked spectrum from the second pathway, and $S_{merge}$ stands for the enhanced speech signal via the merging algorithm. As shown in Equation (8), the proposed merging algorithm utilizes differences and the ReLU function to extract speech components that were commonly missed by the domain-specific networks. Removing these components enhances the overall speech estimation reliability of the proposed speech enhancement model. The merged speech feature vector is transformed into a time-series frame format via the decoder and synthesized into the final objective, a waveform of the speech signal, via the OLA algorithm.

#### 2.3.4. Model Implementation and Training

Figure 4 shows the detailed structure of the implemented dual-path LSTM network.

As shown in Figure 4, the LSTM blocks in the network are composed of layer normalization, LSTM, and fully connected layers aimed at improving the model’s stability and convergence [20]. The depth of the LSTM layers was limited to two to prevent overfitting. The first pathway input involved densely divided frames along the time axis. These frames were encoded using 1D convolution filters with kernel size 16, stride 8, and length 512. The window length for this input was determined considering the system environment, training time, and performance variation across the different window lengths. In this case, a window length of 5 ms with 50% overlap and a window length of 64 ms with 50% overlap were chosen. The second pathway input was created by concatenating the high-resolution magnitude spectrum of the STFT representation and the Mel-spectrum extracted via 128 filter banks. The length of the Mel-filter banks was decided based on the same considerations mentioned earlier.

The LSTM layers were designed with 256 nodes (units). These layers transform semantic time and frequency feature vectors obtained based on the specific feature extraction techniques for each domain into data-structured masks. These masks were estimated accordingly and utilized the same structure as the input data. The estimated masks were multiplied with each input signal to effectively remove the noise components. Then, these masked inputs underwent a restoration process tailored to their corresponding feature extraction techniques that resulted in the generation of speech signals. This architecture and design contribute to the extraction of semantic features from the different domains, which facilitates noise reduction and clean speech restoration of the signals.

## 3. Results

### 3.1. Experiment Environment

All experiments were performed on a system with an Intel(R) Core (TM) I7-10700 CPU @ 2.90 GHz, 16 GB RAM, and GeForce GTX 1660 SUPER 6GB. Data preprocessing, model construction, and training were performed using the TensorFlow and Keras frameworks based on the Python language. Considering the system environment characteristics and the size of the collected dataset, a batch size of 16, a learning rate of 0.001, and 100 epochs were used for model training. Additionally, a learning rate scheduler (early stopping function) was employed to gradually reduce the learning rate by half based on the convergence of the model’s loss value to optimize training time [21].

In the proposed dual-path speech enhancement model, the activation functions used in the noise suppression module responsible for mask estimation are composed of hyperbolic tangent (tanh) and sigmoid functions, depending on each pathway. These activation functions help alleviate the vanishing gradient problem while effectively modeling the nonlinearity of the speech signal, which allows the model to learn more complex and diverse data representations. The non-negative constraint function for the encoder was determined using a grid search, and ReLU was selected as a result. To ensure the model could effectively estimate and remove noise, a loss function based on negative SNR and the Adam (Adaptive Moment Estimation) optimization technique was utilized [22].

We evaluated the performance from the perspective of speech quality and intelligibility using several metrics, including short-time objective intelligibility (STOI), extended STOI (ESTOI), perceptual evaluation of speech quality (PESQ), and signal-to-noise ratio (SNR) [23,24,25,26]. STOI measures the objective intelligibility of a degraded speech signal by correlating the temporal envelopes of the degraded signal with its clean reference. Empirical evidence demonstrates a strong correlation between STOI scores and human speech intelligibility ratings. ESTOI, on the other hand, assesses the objective intelligibility of degraded speech by analyzing spectral correlation coefficients in short time segments without assuming the mutual independence of frequency bands like STOI does. Both STOI and ESTOI scores range from 0 to 1, with higher values indicating better speech intelligibility. PESQ is designed to gauge the subjective quality of perceived speech and produces values ranging from 0 to 4.5. Higher PESQ values correspond to clearer speech. Lastly, SNR quantifies the distortion ratio between clean and enhanced speech. It measures the ratio of the energy of clean speech to the energy of distortion, with higher scores indicating smaller amounts of distortion.

### 3.2. Performance Comparison with Benchmark Models

In this section, the evaluation results of a test dataset comprising 2000 noise-mixed data points are presented. The performance of the dual-path architecture is compared with A1, A2, A3, and the recent benchmark models, DCCRN. These correspond to single-domain models that exclusively utilize complex spectral features, temporal features, Mel-spectral features, and spectral extension blocks, respectively. The results are summarized in Table 1.

As shown in Table 1, our proposed dual-path LSTM model achieves notable enhancements over DCCRN, demonstrating improvements of 0.107 for PESQ, 1.29% for STOI, 2.28% for ESTOI, and 0.33 dB for SNR, under average conditions. These results demonstrate the effectiveness of our dual-domain learning in enhancing speech quality and intelligibility. Considering that the A1 model with time domain features performs better in terms of PESQ and the A2 model with time–frequency domain features performs better in STOI, the proposed method demonstrates consistent and balanced performance across all evaluation metrics. Our proposed method demonstrates a consistent and well-rounded performance across all evaluation metrics. These results support the argument that the dual-path LSTM network appropriately utilizes complementary characteristics to distinguish between speech and noise effectively. Moreover, when comparing ESTOI to STOI, it becomes evident that ESTOI exhibits a more significant degree of improvement. Additional experiments were conducted to confirm whether these results were consistent. Figure 5 shows the STOI and ESTOI evaluation results of DCCRN and the proposed dual-path LSTM.

As shown in Figure 5, the proposed dual-path LSTM model showed an improvement of 9.51% in STOI and 17.51% in ESTOI compared to noisy, and an improvement of 1.61% in STOI and 2.73% in ESTOI compared to DCCRN. Considering Model A3 in Table 1 and the experimental results in Figure 5, the higher improvement rate of ESTOI, which is sensitive to spectral mismatches compared to STOI, supports the effectiveness of the proposed spectral extension block in restoring spectral details that are prone to loss.

Figure 6 illustrates the magnitude spectrum of clean, noisy, and enhanced speech signals, providing a visual comparison between models. As shown in (a) of Figure 6, since DCCRN only considers the spectrum in the time–frequency domain, we can see that there is still residual noise at the beginning and end of the waveform. Notably, the white box section indicates DCCRN’s inability to accurately estimate the spectrum when speech and noise are intertwined. In contrast, the proposed dual-path LSTM network adeptly compensates for the loss of spectral detail, ensuring accurate restoration even under challenging speech and noise conditions. However, looking at the 13–14 s range of the signal improved by the proposed method, it was also found that the clean spectrum was damaged due to some excessive noise suppression. This indicates that multi-domain information may lead to unnecessary suppression of the noise spectrum.

Nonetheless, Figure 6b, which presents the discrepancy between the spectrum of the enhanced signal and the spectrum of clean speech, serves as confirmation that the proposed method is proficient in accurately estimating the speech spectrum across all frequency bands.

## 4. Discussion

The experimental results presented above demonstrate that the proposed dual-path LSTM network not only enhances speech quality but also offers a fresh perspective compared to existing methods that primarily focus on complex spectrum processing. When contrasted with single-domain approaches, whether they are time-based, time–frequency-based, or multi-domain methods, they exhibit superior speech enhancement performance. This improvement can be attributed to two key factors. Firstly, the widely used STFT spectrum in single-domain methods may not provide sufficient information for accurate speech estimation due to trade-offs between time and frequency resolutions. Secondly, time–frequency domain networks, while effective at enhancing the spectrum of noisy speech, may still leave residual noise in the time domain distribution.

To address these issues, we proposed a dual-path LSTM network in which both time and time–frequency domain networks were optimized in a complementary manner along their respective paths. Additionally, ablation experiments underscored the significance of extracting limited-time domain features and the process of compensating for potentially lost spectral details in the speech enhancement procedure.

In future studies, considering the relative simplicity of the proposed network structure, we aim to further enhance its performance via the incorporation of additional structural improvements, such as the integration of attention mechanisms.

## 5. Conclusions

In this paper, we presented a dual-path LSTM network that complemented the advantages of the time and time–frequency domain, considering that single-domain representations such as STFT were difficult to provide sufficient information for speech enhancement. These dual paths conducted LSTM networks to map mask functions, enhancing both temporal and spectral features. To improve the semantic content of temporal features, a 1D convolutional encoder–decoder structure with non-negative constraints was integrated into the time domain path. In the time–frequency domain path, a spectral extension block was introduced to preserve spectral details that were easy to lose. Subsequently, the merging algorithm was applied to combine the enhanced features from each domain, facilitating mutual enhancement and ultimately enhancing the quality of the speech signal. Extensive experimental results showed that the proposed dual-path LSTM network consistently outperformed the existing single-domain methods. The proposed approach is anticipated to serve as an assistive system in speaker recognition and various speech application fields. Additionally, it holds potential for rehabilitation technologies for patients with various hearing impairments in the future.

## Figures and Tables

**Figure 1 bioengineering-10-01325-f001:**
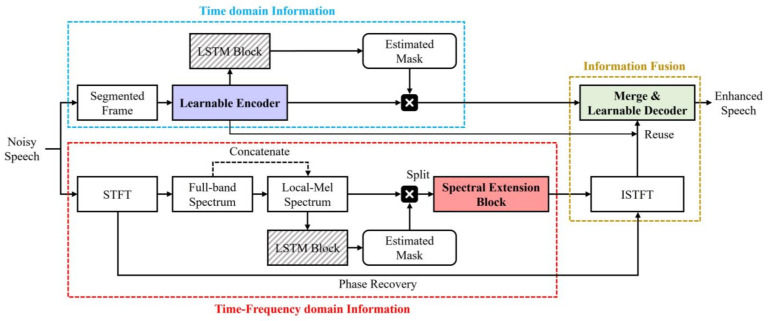
The main architecture of dual-path LSTM-based speech enhancement network.

**Figure 2 bioengineering-10-01325-f002:**
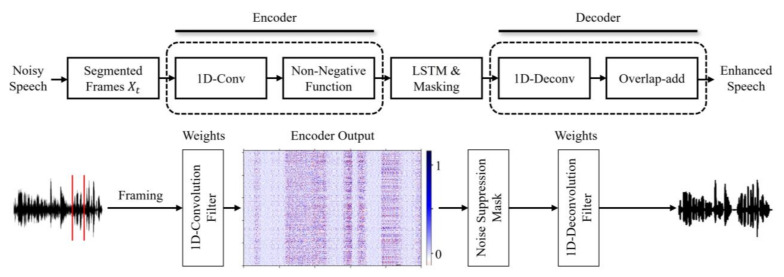
The overall framework of the proposed encoder–decoder architecture to utilize high-resolution time domain features for speech enhancement.

**Figure 3 bioengineering-10-01325-f003:**
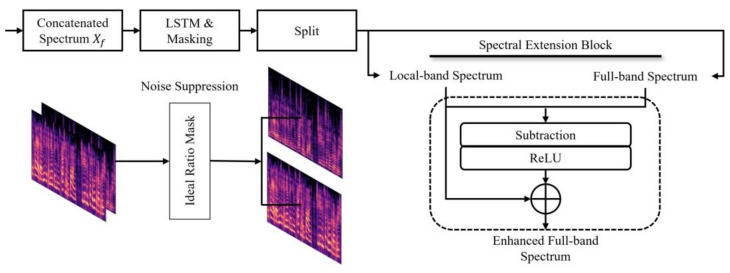
The overall framework of the proposed spectral extension blocks to utilize high-resolution frequency domain features for speech enhancement.

**Figure 4 bioengineering-10-01325-f004:**
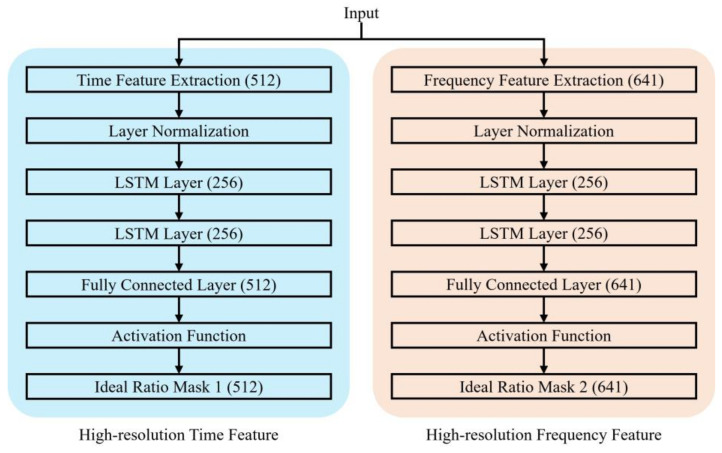
Detailed description of the layers constituting the dual path LSTM network architecture for speech enhancement.

**Figure 5 bioengineering-10-01325-f005:**
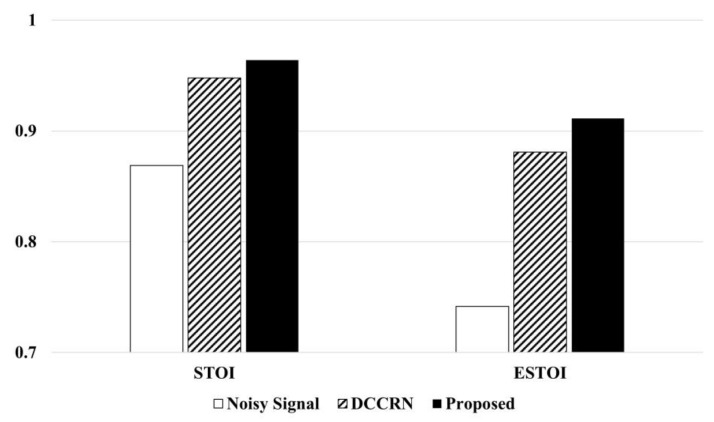
STOI and ESTOI measurement results of benchmark and proposed model.

**Figure 6 bioengineering-10-01325-f006:**
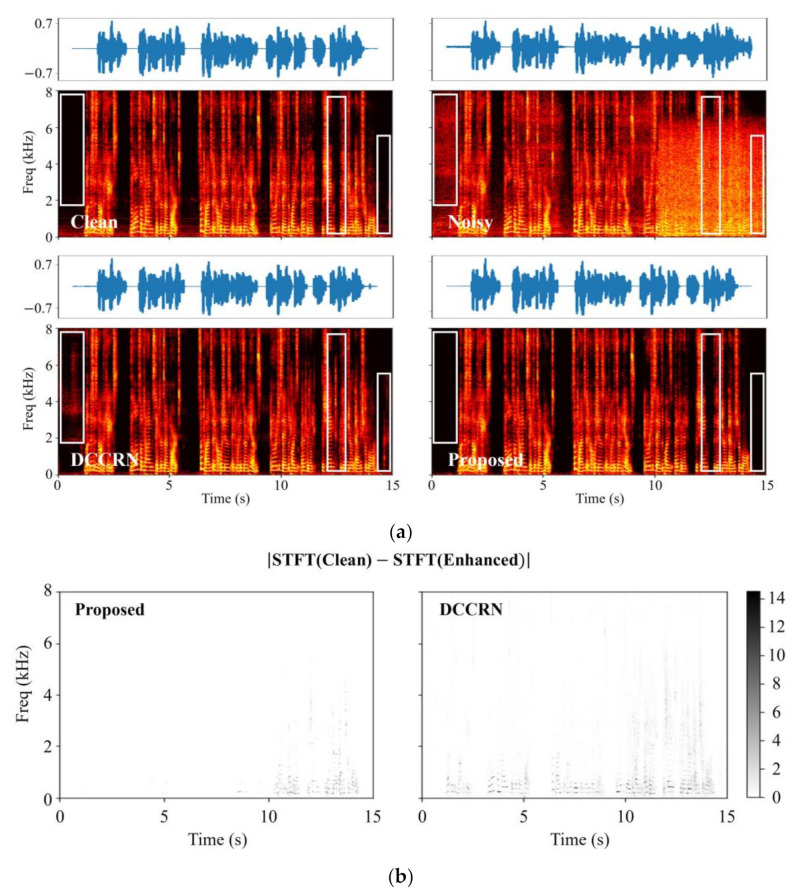
Visualized comparison results of enhanced signals with the proposed model and benchmark model. (**a**) Spectrum and time domain waveforms of clean speech, noise speech, speech enhanced by DCCRN, and speech enhanced by the proposed model. (**b**) Value of difference between enhanced spectrum and clean speech spectrum.

**Table 1 bioengineering-10-01325-t001:** Average value of performance evaluation metrics for each model using 2000 test data.

Model	SNR	PESQ	STOI	ESTOI
DCCRN	16.4456	2.971	0.9395	0.8730
A1 (2 Layer, Non-negative)	14.6096	2.745	0.9206	0.8408
A2 (2 Layer, Mel-Spectrum)	14.7318	2.686	0.9249	0.8522
A3 (2 Layer, Extension Block)	15.0837	2.714	0.9273	0.8673
Dual-path LSTM	16.7756	3.078	0.9524	0.8958

## Data Availability

The references include links to publicly archived datasets.

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
