# Peer review of "Speech Perception Improvement Algorithm Based on a Dual-Path Long Short-Term Memory Network"

_bioengineering, 2023, doi:10.3390/bioengineering10111325_

Round 1
Reviewer 1 Report
Comments and Suggestions for Authors
The paper "Speech Perception Improvement Algorithm based on Dual- 2 Path LSTM Network" presents a novel method to detect speech intelligibility amongst noisy environmental signals. The paper presented some interesting results. There are some however more clarity for the results and discussion are needed.
- The background information provided for the abstract was not directly relevant. For example, the mention of STFT was not needed as this does not seem to be a fair comparison to the method proposed (dual-path LSTM). Please rewrite the abstract to tone down the background information and provide the results and conclusions with possible implications.
- Authors mention utilising metrics for comparisons in the introduction and then in the results. These are important and warrant a descriptive section in the methods to help readers understand the purpose of each metric.
- The results (line 310-311) mentions improvements against noise, but what is the percent improvement against DCCRN
-On the same point, the authors should run some comparison statistics for each of the metrics to see if some of these "improvements" are significant.
-The authors talk about improved speech intelligibility- however, is this really the case? shouldn't speech intelligibility be measured by the number of words that can be clearly heard? How are we certain that the signal did improved speech intelligibility?
- Results and Discussion (especially the discussion section) are too brief. What do the results found mean and how does this compare to other studies. The improvements from DCCRN only seem marginal- are there any advantages for this method if this is the case?
Comments on the Quality of English Language
English seems fine.
Author Response
Dear Dr. Reviewer 1
Thank you for your thoughtful reviewing about this paper. We revised it according to your comment. Your comments were highly insightful and enabled us to greatly improve the quality of our paper. Please refer to the attached file for detailed corrections.

Reviewer 2 Report
Comments and Suggestions for Authors
The paper is a really good one, well written and fluent from the beginning to the end. It deals with a really relevant and modern topic. The paper can be accepted as it is, except for fixing the following:
Citation style sometimes must be improved. Such as: “Hu et al.”, should be “Hu, et al.” and so on.
Moreover, I personally suggest to avoid the use of “we” in scientific writing.
Comments on the Quality of English LanguageI personally suggest to avoid the use of “we” in scientific writing.
Author Response
Dear Dr. Reviewer 2
Thank you for your thoughtful reviewing about this paper. We revised it according to your comment. Your comments were highly insightful and enabled us to greatly improve the quality of our paper. Please refer to the attached file for detailed corrections.

Reviewer 3 Report
Comments and Suggestions for Authors
Speech enhancement is a method aimed at improving speech quality in noisy environments where speech is degraded. The short-time Fourier transform (STFT) analyzes signals in the frequency domain over time and is used in speech enhancement. However, STFT has difficulty obtaining high-resolution information at the same time in the frequency and time domains of a signal.
The AUTHORS present a new speech enhancement model designed with a parallel long short-term memory (LSTM) network structure to identify key speech characteristics in both the time and frequency domains.
THEIR model utilizes a 1-D convolution-based learnable encoder–decoder structure restricted to a non-negative condition to extract time domain high-resolution features and a spectral extension block to combine the sub-band mel and full-band spectrums for effective frequency domain high-resolution features.
THEY highlight that: (1)The dual-path LSTM network estimates time–frequency masks to remove noise, and the masked speech components are reconstructed via the proposed merge algorithm and analog restoration method. (2) The experimental results show that this model outperforms baseline models in various evaluation metrics.
The manuscript is interesting and has merits.
A have some minor comments:
1. The abstract must better summarize each section (e.g. the conclusions)
2. The purpose must be more effective. Avoid the “We” and use the bullet points to detail it better.
3. Describe figures (see for example figure 2-3) in details.
4. Add data labels in in figure 6
5. Discussion and conclusions must be separated. Insert in the discussion the comparison to the literature and the limitations of the study (VERY IMPORTANT)
Author Response
Dear Dr. Reviewer 3
Thank you for your thoughtful reviewing about this paper. We revised it according to your comment. Your comments were highly insightful and enabled us to greatly improve the quality of our paper. Please refer to the attached file for detailed corrections.

Round 2
Reviewer 1 Report
Comments and Suggestions for Authors
The authors have addressed the issues that i have raised.
Comments on the Quality of English LanguageThe discussion and conclusion need to be checked for tense. It should be in the past tense
Author Response
Dear Dr. Reviewer 1
We greatly appreciate your considerate review of our paper, and we have carefully incorporated your valuable feedback in our revisions. Your insightful comments played a pivotal role in enhancing the overall quality of our paper.
